# Comparative Analysis of Water Stress Regimes in Avocado Plants during the Early Development Stage

**DOI:** 10.3390/plants13182660

**Published:** 2024-09-23

**Authors:** Tatiana Rondon, Manuel Guzmán-Hernández, Maria C. Torres-Madronero, Maria Casamitjana, Lucas Cano, July Galeano, Manuel Goez

**Affiliations:** 1Corporación Colombiana de Investigación Agropecuaria (AGROSAVIA), Centro de Investigación La Selva, Rionegro 054040, Colombia; maguzman@agrosavia.co (M.G.-H.); casmi242@hotmail.com (M.C.); lcanog@agrosavia.co (L.C.); 2MRP Laboratory, Research Group on Smart Machine and Pattem Recognition, Department of Electronic and Telecommunications Engineering, Instituto Tecnológico Metropolitano (ITM), Medellín 050034, Colombia; mctorresm@unal.edu.co (M.C.T.-M.); manuelgoez@itm.edu.co (M.G.); 3Department of Computer and Decision Sciences, Faculty of Mines, Universidad Nacional de Colombia, Medellín 050034, Colombia; 4Laboratori d’Anàlisi i Gestió del Paisatge, Universitat de Girona, 17071 Girona, Spain; 5Research Group Materiales Avanzados y Energía MatyEr, Instituto Tecnológico Metropolitano (ITM), Medellín 050013, Colombia; julygaleano@itm.edu.co

**Keywords:** avocado seedling, phenotyping, spectral images, vegetation index, water stress

## Abstract

The avocado cv. Hass requires a suitable rootstock for optimal development under water stress. This study evaluated the performance of two avocado rootstocks (ANRR88 and ANGI52) grafted onto cv. Hass under four water stress conditions, 50% and 25% deficit, and 50% and 25% excess during the nursery stage. Plant height, leaf area (LA), dry matter (DM), and Carbon (OC) content in the roots, stems, and leaves were measured. Root traits were evaluated using digital imaging, and three vegetation indices (NDVI, CI_RE_, and MTCI) were used to quantify stress. The results showed that genotype significantly influenced the response to water stress. ANRR88 exhibited adaptation to moderate to high water deficits. ANGI52 adapted better to both water deficit and excess, and showed greater root exploration. LA and DM reductions of up to 60% were observed in ANRR88, suggesting a higher sensitivity to extreme changes in water availability. More than 90% of the total OC accumulation was observed in the stem and roots. The NDVI and the MTCI quantified the presence and levels of stress applied, and the 720 nm band provided high precision and speed for detecting stress. These insights are crucial for selecting rootstocks that ensure optimal performance under varying water availability, enhancing productivity and sustainability.

## 1. Introduction

Over the last 20 years, the international avocado (*Persea americana* Mill.) trade has grown from USD 331 million to USD 7.59 billion, and the United States and some European countries, such as the Netherlands, Spain, and France, lead the importation markets [1]. Avocados have a huge market demand due to their highly nutrition-dense compounds and benefits to human health [2,3,4]. The avocado cv. Hass is the most popular variety of avocado worldwide, and its consumption has tripled since the early 2000s [3,5]. In the United States, per capita consumption has increased more than 300% since 2000, from 1.02 to 4.01 kg person^−1^ yr^−1^ [6,7].

Colombia is included in the top five avocado cv. Hass producers and exporters worldwide, with a high expansion rate of fresh fruit suppliers to Europe and the United States. In Colombia, the Hass variety represents 26% of the planted area, equivalent to 20.000 ha, approx. [8]. Moreover, it is cultivated in different ecosystems and at different technological levels [9]; mainly in the Andean’s mid-altitude ranges from 1100 to 2250 m above sea level. Therefore, it is essential to study the behavior of this plant species under different stress conditions.

Climate variables are shifting as a result of global warming, making Colombia particularly vulnerable to its effects. A variable precipitation pattern, temperature, and carbon dioxide (CO_2_) concentration increases, and air humidity variation constantly change and affect plant water use [10]. In addition, population growth, land use change, and increasing demand in non-agricultural sectors impact the availability and quality of water [11,12].

Freshwater resources are increasingly over-exploited in many parts of the world [13]. In regions of the world where avocados are cultivated, such as the Mediterranean region and southeast California, drier conditions and water stress are on the increase due to climate change, and fluctuations in these factors directly affect crop development [14,15]. Meanwhile, Colombia has significantly increased its agricultural water consumption over the past two decades, rising from 4.92 to 25.04 km^3^ year^−1^ [16,17], mainly due to the use of empirical irrigation practices by farmers and the lack of water requirement information.

Avocado plants are susceptible to events associated with water stress [18,19], which can significantly impact fruit production. Both water deficits and excesses can induce alterations in the plant and its physiological processes, primarily due to its sensitive shallow root system, affecting shoot and root growth, flowering time, and biomass production [20,21,22]. The duration and severity of water stress are other relevant factors when stress responses are analyzed, and several studies have explained the mechanisms plants use to cope with water stress [14,15,23]. Complex physiological and biochemical adaptations are developed by the avocado plants in response to the stress level applied, which results in fruit drop during formation and decreases in fruit weight and quality [24,25,26]. 

Avocado productivity and fruit quality decrease when the plants are grown in soils with high water retention [27], as this promotes root damage by anoxia [28]. Understanding the effect of abiotic stress on the roots of a permanent crop, such as avocado, is essential to guarantee the sustainability of the productive system, since it allows us to know the “tolerance capacity” of plant materials to stress conditions directly and indirectly, which allows for the selection of rootstocks suitable to specific production areas. Anchoring and support in the soil, water, nutrient assimilation, and a beneficial relationship with the soil microbiota are some of the main functions of the roots, which are in direct contact with soil and can quickly detect changes or alterations caused by abiotic stresses, such as drought, waterlogging, and salinity, among other factors [29,30].

Plant phenotyping is a powerful tool for studying the water stress that significantly impacts avocado productivity. In this regard, several studies in morphological [31,32], physiological [26,33,34], biochemical [35,36], and molecular [37,38,39] traits were reported to assess the responses of avocado plants under water stress.

Recently, advanced imaging technologies, such as thermal, multispectral, and hyperspectral cameras, have enabled the precise measurement of traits [40,41,42]. These non-invasive and non-destructive measurements help to identify early stress responses and to quantify traits more efficiently.

Commercial avocado orchards use asexual propagation via grafting. This method is the most recommended, and is used by farmers and academics. It involves a commercial cultivar as a scion and usually a local landrace as a rootstock. The use of grafted rootstocks is ideal for mitigating root pathogen effects [43,44,45] or abiotic stresses [46,47]. In this study, we used two experimental avocado rootstocks that show potential for use under water stress.

Previous studies have provided evidence of the avocado rootstock effect on the performance of cv. Hass. However, a more in-depth evaluation and quantification of the biometrical, physiological, and spectral responses of both rootstock and scion during the initial growth stages are required to guarantee optimal plant development. The aim of this study was to assess the performance of two creole avocado rootstocks grafted onto cv. Hass under different water stress conditions during the early growth stage. This evaluation was important to identify rootstocks suitable for specific water regimes, and contributes to research focused on physiological and omics approaches, as well as water management practices.

## 2. Results

### 2.1. Root Analyses

An analysis of variance was performed for root traits (Table 1). Statistical differences (*p* ≤ 0.05) were observed between rootstock effects for the size of the entire root system (AREA), total root length (TRL), right root angle (RRA), and root angle opening (RAO). Among the different water regimes evaluated, statistical differences (*p* ≤ 0.05) were observed in the following traits: depth with a maximum proportion of roots (DPR), AREA, TRL, left root angle (LRA), and RRA. For the effects of rootstock–water regime interactions, TRL, LRA, RRA, and RAO were statistically different at the 0.05 level.

DPR ranged from 15.24 to 31.77 cm (Table 2). The lower value of DPR for genotype ANRR88 was observed in T4 (50% of water excess) with 15.24 cm, while for ANGI52, it was 22.94 cm in T3 (25% of water excess). Overall, a higher root exploration was observed in ANGI52 across all the treatments, with a mean value of 25.58 cm. AREA represents the size of the entire root system, expressed in cm^2^. Between genotypes, AREA was higher for ANGI52 than ANRR88, with mean values of 264.3 cm^2^ and 339.6 cm^2^, respectively. The minimum root exploration in the soil was observed for ANRR88 under treatment T4 (50% of water excess) at 190.09 cm^2^. However, the maximum AREA was observed in ANGI52 under treatment T3 (25% of water stress), with a value of 397.23 cm^2^.

For both genotypes, the treatment values of TRL were lower than those observed for the control treatment (T5). For ANRR88, a reduction of 18% on average was observed for TRL compared with T5, making it the treatment with the smallest reduction in root length. A higher decrease in TRL values was observed for ANGI52, particularly for treatments T1 (50% of water restriction) and T2 (25% of water restriction), with values of 1680.4 and 1560.5 mm, respectively.

LRA and RRA indicate the outermost angle to the horizontal along an arc of 10 cm. Root angle opening (RAO) is the opening of the angle between LRA and RRA. In both genotypes, T5 showed the maximum values of LRA and RRA among the treatments evaluated. For the genotype ANRR88, mean opening angles of 65.1° and 64.7° were observed for LRA and RRA, respectively. The lowest value of LRA was observed in T1 (49.0°), while the lowest value of RRA was observed in T3 (32.8°). Similar values were observed for ANGI52, with mean values of 64.5° and 66.4° for LRA and RRA, respectively. Maximum values of RAO were observed for water restriction treatments in T1 (61.5°) for ANRR88, and in T3 (63.3°) for ANGI52.

The results shown below in Section 2.2 validate the selection of genotypes ANRR88 and ANGI52 based on the results of the root phenotyping. The information provided by the variables plant height, leaf area, dry matter, and organic Carbon per organ (root, stem, and leaf) allowed us to know the effect of water stress on the entire avocado plant, from the roots, represented by ANRR88 and ANGI52, to the aerial part, represented by the cv. Hass.

### 2.2. Plant Height, Leaf Area, Dry Matter Accumulation, and Organic Carbon Content

Averaged data across plots were used to conduct an analysis of variance for plant height (PH), leaf area (LA), dry matter accumulation, and organic Carbon (OC) content for leaves, stems, and roots (Table 3). No significant differences were observed between rootstocks (RS) for these traits. However, water stress regimes (WR) had a significant effect on all traits. The interaction RS × WR effect was significant only for stem dry matter at a significance level of 0.05.

Plant height (PH) showed statistically significant differences (*p* ≤ 0.05) among the treatments, but not between the genotypes evaluated. Genotype ANRR88 showed more height at 61.68 cm (±1.32), while ANGI52 reached 60.24 cm (±1.37). For both genotypes, the treatment with the highest PH was T5 with an average of 69.24 cm (±2.48), while the lowest PH was observed in T1, with 55.40 cm (±2.21) (Figure 1). The PH variable decreased because of the treatments. In cases of severe stress due to water deficit or excess, the lowest values were recorded, 55.40 cm (±2.21) and 56.67 (±2.02) cm, respectively.

Highly significant differences were found for leaf area (LA) under the water stress treatments evaluated for the genotypes. ANGI52 presented the lowest values for this variable, with 650 cm^2^. For leaf area (LA), statistically significant differences (*p* ≤ 0.05) were observed among genotypes and treatment effects (Figure 2). The control (T5) showed a mean value for rootstock of 3053.2 cm^2^, superior to all the treatments evaluated. A higher reduction in LA was observed for ANRR88 compared with ANGI52, ranging from 12.2% (T1) to 65.7% (T2). A higher LA reduction was observed for ANRR88 in T4 when compared to the other treatments, with a value of 711.1 cm^2^. The highest value of LA was observed in ANGI52 for T2, 2204.5 cm^2^.

The dry matter accumulation was quantified to leaves, stems, and roots (Figure 3). In both rootstocks, the control treatment (T5) showed higher values of dry matter accumulation, ranging from 1396.89 (ANGI52) to 1443.79 g m^−2^ (ANRR88). A higher variation in dry matter accumulation among treatments and organs was observed in ANRR88, where T4 (50% of water excess) showed a maximum reduction of 57.3% in dry matter accumulation compared with the control treatment (T5). However, when the same treatment (T4) was applied to the rootstock ANGI52, a smaller reduction in dry matter accumulation was observed, with a decrease of 38% compared with T5 (Figure 3).

The leaves were the plant organ with the highest decrease in dry matter, ranging from 44.1% (ANGI52) to 63.3% (ANRR88). A maximum reduction in stem and root dry matter was observed for ANRR88, 44.1 and 46.3%, respectively.

The organic Carbon (OC) content was determined for both rootstocks and in each plant organ (Figure 4). ANRR88 accumulated a higher Carbon content amount than ANGI52 in all plant organs, with average values of 10.94 and 4.66 mg OC g^−1^ of DT, respectively. At the plant organ dissertation level, high Carbon accumulation was observed in the stem and roots, where both organs accumulated more than 90% of the total Carbon determined. However, a higher Carbon accumulation proportion in the stem was observed for ANRR88, ranging from 68.1 to 88.4%, for all the treatments.

T3 caused a lower accumulation of OC in roots for ANRR88; even so, all the treatments, due to a deficit or excess of water, strongly affected the accumulation of OC in the roots, while ANGI52 was affected by T1 (Figure 4). In both genotypes, the highest concentration of OC in the stem was in T5. ANRR88 and ANGI52 had the lowest values of OC in the stem under T1 with 3.10 and 0.27 mg of OC g^−1^ DT, respectively. The OC content in leaves was higher in T4 for both genotypes.

### 2.3. Analysis of Vegetation Indices

The vegetation indices were evaluated as the mean values across both rootstocks, and the results are presented per treatment (Table 4). Statistical differences (*p* > 0.05) were observed among the treatments for the three vegetation indices evaluated. The coefficient of variation ranged between 1.09% and 16.72% for the normalized difference vegetation index (NDVI) and the MERIS Terrestrial chlorophyll index (MTCI), respectively.

Figure 5 summarizes the mean and standard deviation for the NDVI, the CI_RE_ index, and the MTCI. For the NDVI, differences were observed among treatments, with T5 (control) showing a higher value (0.9318) compared to the other treatments. T2 was the treatment that most significantly affected the rootstocks according to the NDVI values. With the data obtained in our experiments, T1, T3, and T4 could not be statistically differentiated from each other. The MTCI found a significant difference between T3 and the rest of the treatments. T1, T2, and T4 were not statistically different (*p* ≤ 0.05). T2 was the treatment with higher MTCI values of 5.8324. Finally, the CI_RE_ index indicated that T3 significantly differed from the other treatments evaluated. 

The NDVI and the MTCI were more effective than the CI_RE_ index at quantifying stress, distinguishing not only between water excess and deficit stress, but also between different levels of these stresses. Stress detection was achieved using all vegetative indices including the NDVI, the CI_RE_ index, and the MTCI.

## 3. Discussion

This study evaluated the responses of two avocado rootstocks, ANRR88 and ANGI52, under different water regimes, analyzing both root and plant phenotypic traits and vegetation indices. The results provide valuable insight into how each rootstock responds to conditions of water deficit or excess, which is crucial for the identification and selection of rootstocks, improving crop management practices under variable nursery and field conditions.

Several techniques and software were used for crop root phenotyping, increasing the accuracy and timing of the data acquisition [48,49,50]. Digital imaging utilization for avocado root phenotyping allows for the quantification of root traits like architecture, root growth dynamics, and root interaction with the soil. However, there are no published studies on avocado root phenotyping; it is costly in terms of resources and time, limiting factors in the research programs. The REST v1.0.1 software was used in this study to facilitate the acquisition and data processing of several root traits. The accuracy of this software was previously validated by [50,51,52], which found a high correlation between manually measured traits and image-derived traits.

The statistical differences observed in root traits (DPR, AREA, TRL, LRA, RRA, and RAO) among the different water regimes evaluated, using the REST software, suggest a significant phenotypic plasticity in response to soil condition. For each treatment applied, ANGI52 showed greater root exploration and root size systems (AREA) compared to ANRR88 (Table 2), indicating better anchoring, and water and nutrient uptake capacity under water excess conditions (T3). For the RAO, root behavior varied depending on the stress applied. Severe water stress to avocado plants changes root architecture, resulting in both a larger root angle opening and a higher proportion of shallow roots.

The difference in plant height (PH) and leaf area (LA) between rootstocks and treatments indicates that water deficit and excess differently affect the vegetative growth of the avocado plant. Water stress inhibits cell division and growth, as well as some developmental processes of the plants [53]. The reduction in LA was more significant in ANRR88 than in ANGI52, suggesting ANRR88 has a higher sensitivity to extreme changes in water availability.

Dry matter partitioning was affected when the water regime was modified, particularly in the roots and stem. The dry matter accumulation in ANGI52 was similar among treatments for each plant organ evaluated. However, a significant reduction was observed in T1, with a decrease of 43% compared to the control treatment. Several authors have explained that the variation in dry matter partitioning and accumulation is caused both by a competitive growth under limitations of assimilates and by hormone regulation [54,55,56]. PH, LA, and dry matter accumulation are closely correlated with crop productivity [57]. These results suggest the need for complete differential soil management according to the genotype used as rootstock, and special attention to the substrate composition such as soil texture, in the case of field conditions and nursery, respectively.

Variations in organic Carbon accumulation among plant organs and genotypes were observed. ANRR88 was highly affected in terms of OC content by treatments associated with water restriction, T1 and T2, compared with the water excess treatments. Meanwhile, ANGI52 showed low OC accumulation but similar content per organ among the treatments. In plants subjected to abiotic stress, photosynthetic Carbon uptake is limited [58], growth decreases or stops, and non-structural carbohydrates (e.g., glucose, fructose, sucrose, and starch) are accumulated [59,60]. When Carbon is stored instead of being used to increase dry matter, there is a notable reduction in light capture and photosynthetic capacity. In our study, more than 90% of Carbon accumulation was observed in both stem and roots. Root architecture, whether deep or shallow, affects Carbon stocks, resulting in a higher adaptation capability to environmental changes [61,62]. Plant storage accumulation starts as soon as abiotic stress is detected, leading to the accumulation of photoassimilates in lower organs, such as the stem and roots, while the leaves reduce their photosynthetic area to avoid excess evapotranspiration. In this study, this indirect relationship was observed to be overlapping in Figure 3 and Figure 5.

Water restriction has a progressive effect on plants, causing a distinct response that eventually results in reduced leaf expansion and increased root growth. Other effects of water stress are the alterations in gene expression, particularly in relation to the synthesis of enzymes [63]. 

Rocha-Arroyo et al. [64] indicate that root development occurs in phases opposite to vegetative shoots development. Both processes can be affected in the presence of water stress, as evidenced by the results showing reduced leaf development under stress conditions.

Water is not the only stress that affects the normal physiological behavior of plants. Variations in temperature, light, and atmospheric CO_2_ concentration impact the distribution of photoassimilates and the generation of biomass [28,65].

In orange trees, another grafted fruit tree, Garzón et al. [66] showed that water deficit stress can impact fruit setting in citrus cv. Valencia. However, when the stress is removed after the first flowering, the plant can increase its fruit set efficiency compared with plants that continue to experience water deficit stress after flowering. Since the greatest proportion of vegetative development occurs when there is an adequate water supply and stops when there is not, once produced, the vegetative structures compete with the fruits for OC.

The leaf is the organ of the plant that exhibits the greatest negative water potential when excessive transpiration occurs, and the plant vigor can primarily be impacted in the leaves during water deficiency stress [67]. The spectral response of the leaves can also change in response to the type of stress and its intensity, as evidenced by comparing different vegetation indices (NDVI, MCTI, and CI_RE_). The obtention of precise wavelength ranges by type of stress, as identified in this study, allows for early intervention in several aspects: (i) remediating stress, (ii) early selection of plant materials with tolerance to specific stresses, (iii) selecting efficient predictive variables, and (iv) designing stress prediction models, among others. 

The spectral response to mild level and short-term stresses may not be as expected. Plants have different mechanisms to cope with stress, such as decreasing the water potential of the tissues, such as osmotic adjustment, which maintains turgor in the cell wall, and is reflected as “vigor” [68]. Additionally, mechanisms such as gene expression can cause some genes to be turned off in the presence of excess stress and others to be activated by deficit stress [63]. This could alter both the spectral response and the accumulation of biomass and organic Carbon.

The metric that best quantified stress among treatments in this investigation was the CI_RE_ index, which is based on the amount of chlorophyll between the NIR bands and the red edge, specifically between 700 nm and 1000 nm in wavelength. We suggest that simultaneous spectral alterations occur in multiple electromagnetic regions, with the most remarkable changes occurring in the infrared region. The reflectance in this region is attributed to cell structure and water content due to the intricate interaction between the reflectance of the cavities between the leaf and the internal reflection of infrared radiation from these cavities [69].

Currently, our research program involves the constant monitoring of the rootstocks under both controlled and field conditions. Data from physiological traits, genetic characterization, and spectral (multi and hyperspectral) imaging are being processed for an integration analysis in the next phase. Different artificial intelligence (AI)-based methods such as machine learning (ML) and deep learning techniques (e.g., convolutional neural networks—CNNs) have been used preliminarily for the efficient processing of large and complex datasets. These approaches will be applied to model the rootstocks and scion responses to varying water availability by identifying key stress markers. The use of ML and CNN techniques in the identification, classification, and prediction of water stress in crops is rapidly increasing, making their application both feasible and highly promising.

## 4. Materials and Methods

### 4.1. Genetic Material and Experimental Conditions

This study used two experimental avocado rootstocks, ANRR88 (Genotype 1) and ANGI52 (Genotype 2). Both rootstocks were selected previously by AGROSAVIA from local landraces of Antioquia, a northwest department of Colombia. These landraces were collected and evaluated for their agronomic performance and genetic structure in previous studies [70,71,72]. Seeds from selected avocado trees were germinated in a nursery, and after three months, those with uniform sizes were grafted with cv. Hass scions. Grafted plants were grown in a greenhouse (19 ± 3 °C, 72 ± 4% RH) for five months. They were transplanted to buckets of 20 L capacity filled with a commercial substrate. Chemical and physical analyses of the substrate (Appendix A) were carried out before the installation of the experiment to determine the initial properties of the substrate and adjust the fertilization doses administered for plant nutrition. The substrate used in this study shares soil properties similar to those in the region. Ten months after grafting, healthy and vigorous plants were selected for the experiment. Seedlings and grafted plants were irrigated twice weekly to ensure the correct development of the plants. 

The experiment was performed in 2021, in a greenhouse of AGROSAVIA, located at La Selva Research Center, Colombia (6°7′46.5″ N, 75°24′55.6″ W, 2110 m.a.s.l). Based on the average of 40 years of meteorological data, the climate in the area featured an average annual temperature of 17.8 °C, precipitation of 1917 mm, relative humidity of 78%, daylight duration of 1726 h yr^−1^, and evapotranspiration of 1202 mm.

### 4.2. Treatments and Experimental Design

Grafted plants were exposed to five water levels: a control treatment (T5) with non-stressing water conditions, which consisted of a constant water application of 222 mL day^−1^ plant^−1^ according to Dorado-Guerra et al. [73] for the volume of soil used; a water restriction treatment of 50% (T1) and 25% (T2) less water than the control amount, with a twice a week watering regime; and a water excess treatment with 25% (T3) and 50% (T4) more water than the control amount, with a daily irrigation regime.

All treatments were laid out in a split-plot arrangement under a randomized complete block design with nine replications, where the whole plot and sub-plots were rootstocks and water regimes, respectively, resulting in a total of 90 plots, where each plant represented each experimental unit.

### 4.3. Trial Management

Prior to the experiment, the substrate was saturated with water and drained for 24 h. Tensiometers were placed in each bucket to monitor daily soil moisture and water content at 10 cm depth during the trial. Soil water drainage was ensured through holes at the bottom of the buckets. Treatments were maintained for 180 days until non-destructive and destructive sampling.

The plants received the recommended fertilization rates of 87.7 g plant^−1^ of nitrogen (N), 64.4 g plant^−1^ of phosphate (P_2_O_5_), and 84.7 g plant^−1^ of potassium (K_2_O) per bucket. The fertilization was divided and applied as follows: 50% one week before the experiment started, 25% 30 days after experiment started, and the remaining 25% 90 days after the experiment started.

### 4.4. Data Collection

Soil moisture per bucket was monitored at 10-day intervals (n = 3 per plot) using a FieldScout^®^ TDR 300 (Spectrum^®^ Technologies Inc., Aurora, IL, USA) and expressed as a percentage (%). At 180 days, plants were sampled for both non-destructive and destructive assessments in each plot. Plant height (PH, cm) was measured on the main stem from the ground level to the canopy terminal. Entire plants were harvested and divided into leaves, stems, and roots. Leaf area (cm^2^) was measured for all the leaves harvested from each plot and scanned in triplicate using a LI-3000C (LI-COR^®^ Bioscience, Lincoln, NE, USA) portable leaf area meter.

Roots were extracted one at a time to avoid moisture loss and changes in the architecture of the roots before imaging. The roots were carefully shaken briefly and washed with regular water to remove the remaining soil. Following the method described by Colombi et al. [52], the roots were photographed in a custom-made imaging tent with modifications. Plastic tubes for the structure and a combination of cold and warm white regular LED panel lights were used in place of aluminum bars and flashlights, respectively. A 24.2 megapixel Nikon D5500 digital camera (Nikon^®^, Tokyo, Japan) with a 35 mm fixed focal objective was used to photograph the root architecture. The camera was mounted on a tripod and placed in the center of the tent, 70 cm above the ground, and at a fixed distance of 100 cm from the roots. Uniform distance from the camera to roots, exposure time, and illumination settings were fixed to normalize and optimize image quality. A minimum of 25 photographs were taken per root sample. Images were recorded, stored in JPEG format, and immediately renamed with an ID code. The details of the root capture, image acquisition, and analysis of avocado roots subjected to water stress are shown in Figure 6.

The quantitative traits of the root system architecture were obtained from the images. Depth with a maximum proportion of roots (DPR, cm) was estimated. The size of the entire root system (AREA, cm^2^) was calculated as the sum of all pixels within the convex hull. Total root length (TRL, mm) was the total projected structure length within the convex hull. The left (LRA, °) and right root angle (RRA, °) were determined as the outermost angle to the horizontal along an arc of 10 cm. The root angle opening (RAO, °) was determined as the opening angle between LRA and RRA.

Plant organs (roots, stems, and leaves) were assessed for estimated dry matter production and organic Carbon (OC). Samples were air-dried for 120 h and further dried using an electric stove at 45 °C until constant weight for dry matter was attained. OC was estimated for each organ, using wet digestion with the dichromate method [74] in triplicate.

### 4.5. Hyperspectral Image Acquisition and Vegetation Index

A hyperspectral image was captured on 18 November 2021 using a Dual HySpex Mjolnir VS-620 (HySpex^®^, Oslo, Norway) system. At the time of imaging, the plants had been under stress for 155 days. The system included a V-1240 (HySpex^®^, Oslo, Norway) sensor with a spectral range between 400 and 1000 nm and comprising 200 spectral bands with a bandwidth of 3.0 nm. The image was captured by a drone at an altitude of 2140 m.a.s.l, approximately 40 m from the surface. The images were orthorectified and atmospherically corrected. Manual labeling was performed to match each section of pixels corresponding to each avocado rootstock and treatment. From these spectral signatures, three vegetation indices were calculated and subjected to statistical analysis: normalized difference vegetation index (NDVI), Chlorophyll estimation red edge (CI_RE_), and the MERIS Terrestrial chlorophyll index (MTCI). These indices were selected for their sensitivity to chlorophyll and irrigation levels [75]:(1)$$NDVI\;=\;\frac{R_{NIR}-R_{R}}{R_{NIR}+R_{R}}$$
(2)$$MTCI=\frac{R_{NIR}-R_{RE}}{R_{RE}-R_{R}}$$
(3)$$CI_{RE}=\frac{R_{NIR}}{R_{R}}-1$$
where R_NIR_, R_RE_, and R_R_ correspond to the reflectance in the near-infrared (780 to 900 nm), red edge (690 to 710 nm), and red (650 to 680 nm), respectively.

### 4.6. Data Analysis and Statistics

For root image analyses, the Root Estimator for Shovelomics Traits (REST, v1.0.1 Institute of Agricultural Science, ETH, Zurich, Switzerland) software was utilized in MatLab 7.12 (The MathWorks, Inc., Natick, MA, USA). The image pre-processing process was realized as described by [76]. 

A two-way ANOVA was performed for each dependent variable, with rootstocks and water regimes treated as fixed effects in the model. Where significant effects were observed, paired comparisons were conducted using the Tukey test at a significance level of *p* ≤ 0.05. The normality assumption was confirmed based on the Shapiro–Wilk test. All data, including those from the REST software, were analyzed in R v4.1.1 (R Core Team 2021).

## 5. Conclusions

This study allowed us to characterize the behavior of two avocado rootstocks during the nursery phase under abiotic stress, such as water stress. The response to water stress was conditioned by the genotype. ANRR88 presents a greater ability to adapt to stress conditions due to moderate to high water deficit for medium-term periods, while ANGI52, shows better adaptation to both deficit and excess stress conditions, with a larger root system and greater root length. Both genotypes showed significant reductions in plant height and leaf area under severe water stress. The leaves experienced the highest decrease in dry matter accumulation under water stress. More than 90% of the total Carbon was stored in the stem and roots, with ANRR88 showing a higher proportion of Carbon accumulation in the stem across all treatments.

Water stress affected the spectral responses of avocado plants. The spectral bands near the red edge, around 720 nm, were shown to be more efficient in quantifying this stress. These spectral tools support faster measurements with high precision and in a shorter time than conventional techniques require. All vegetation indices used in this study detected stress, but their effectiveness varied in distinguishing among different levels of stress. The MTCI and the NDVI were more effective for detecting and quantifying stress under varying water conditions.

These findings enhance our understanding of how avocado plants respond to water stress, using simple and accessible tools that are highly precise and effective in detecting and quantifying stress. This knowledge will allow for improvements in future water management practices by aligning them with the plant’s natural stress regulation strategies.

Further investigations, including additional physiological variables and AI-based approaches, are necessary to elucidate the mechanism by which water deficit or excess influences the behavior and response of the rootstock–scion interaction. 

## Figures and Tables

**Figure 1 plants-13-02660-f001:**
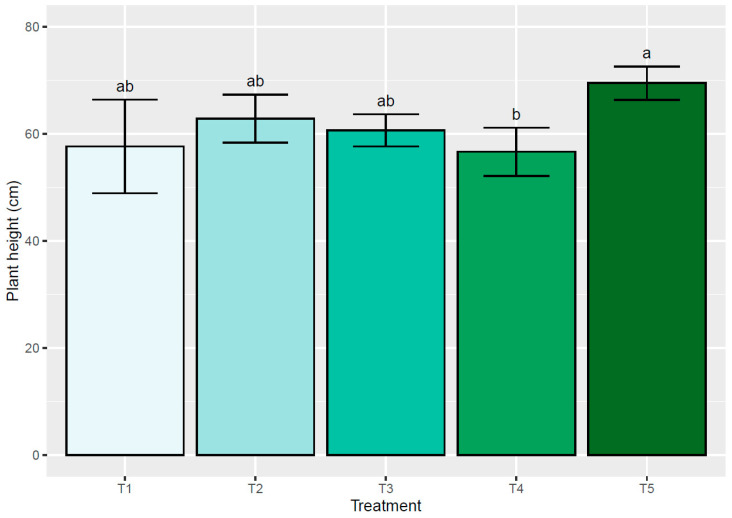
Average plant height (cm) for the different treatments evaluated: water restriction of (T1) 50% and (T2) 25%, water excess of (T3) 25% and (T4) 50%, and (T5) control. Values represent the mean of the two rootstocks evaluated. Both rootstocks were combined with a common cv. Hass scion. Different letters indicate statistical differences at a significance of *p* ≤ 0.05 according to the Tukey test.

**Figure 2 plants-13-02660-f002:**
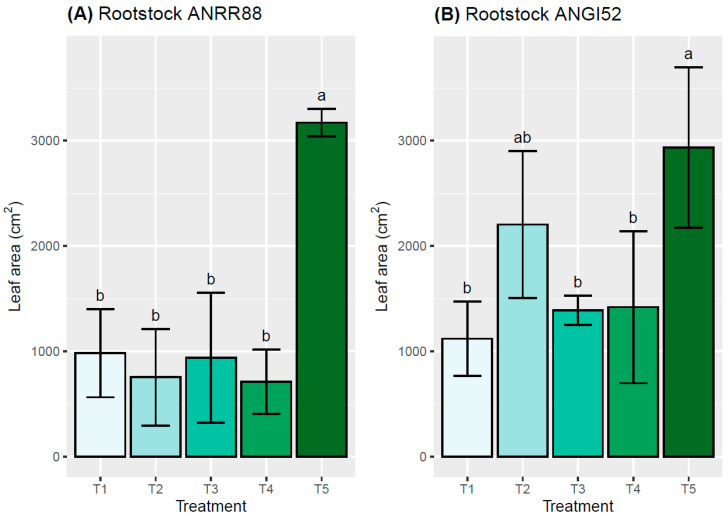
Leaf area (cm^2^) of the avocado plants using rootstock (**A**) ANRR88 and (**B**) ANGI52 for the different treatments evaluated: water restriction of (T1) 50% and (T2) 25%, water excess of (T3) 25% and (T4) 50%, and (T5) control. Both rootstocks were combined with a common cv. Hass scion. Different letters indicate statistical differences at a significance of *p* ≤ 0.05 according to the Tukey test.

**Figure 3 plants-13-02660-f003:**
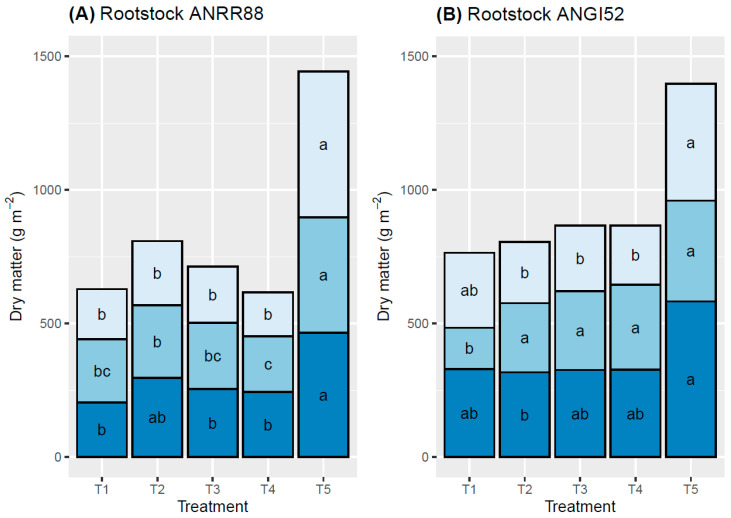
Dry matter (g m^–2^) partitioning per plant organ: leaves (light blue), stem (medium blue), and roots (dark blue) for ANRR88 (**A**) and ANGI52 (**B**). Treatments: water restriction of (T1) 50% and (T2) 25%, water excess of (T3) 25% and (T4) 50%, and (T5) control. Both rootstocks were combined with a common cv. Hass scion. Different letters indicate statistical differences at a significance of *p* ≤ 0.05 according to the Tukey test.

**Figure 4 plants-13-02660-f004:**
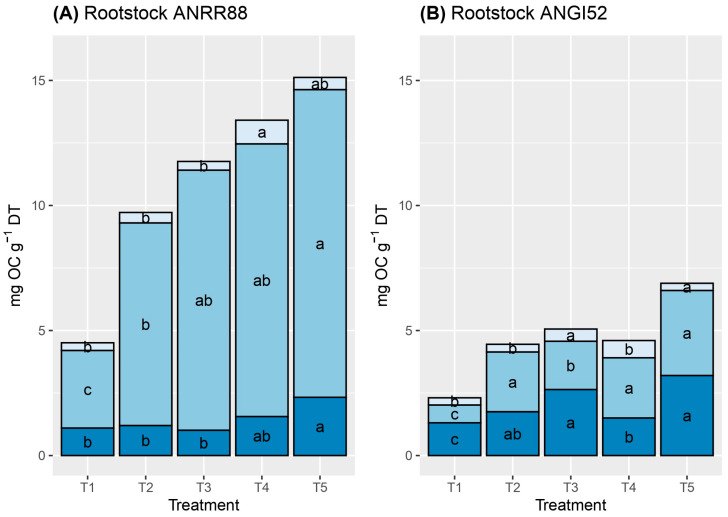
Organic Carbon (mg OC g^–1^ DT) partitioning per plant organ: leaves (light blue), stem (medium blue), and roots (dark blue) for ANRR88 (**A**) and ANGI52 (**B**). Treatments: water restriction of (T1) 50% and (T2) 25%, water excess of (T3) 25% and (T4) 50%, and (T5) control. Both rootstocks were combined with a common cv. Hass scion. Different letters indicate statistical differences at a significance at the *p* ≤ 0.05 according to the Tukey test.

**Figure 5 plants-13-02660-f005:**
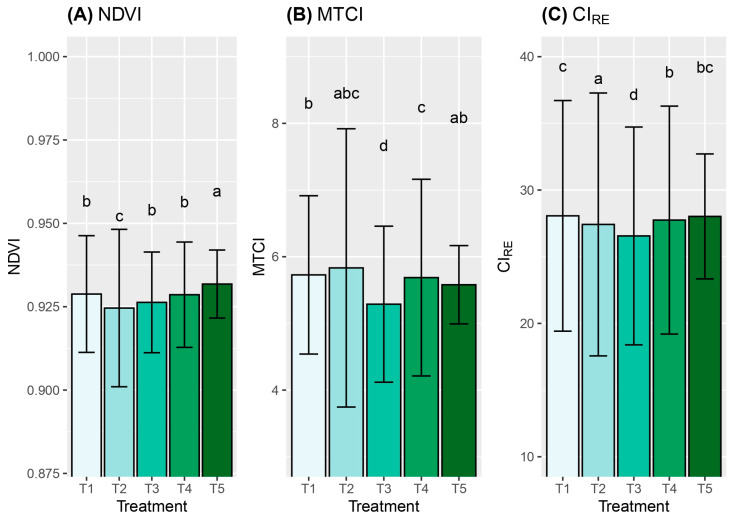
Mean values per treatment for (**A**) NDVI, (**B**) MTCI, and (**C**) CI_RE_. Treatments: water restriction of (T1) 50% and (T2) 25%, water excess of (T3) 25% and (T4) 50%, and (T5) control. Both rootstocks were combined with a common cv. Hass scion. Different letters indicate statistical differences at a significance of 0.05 level, according to the Tukey test.

**Figure 6 plants-13-02660-f006:**
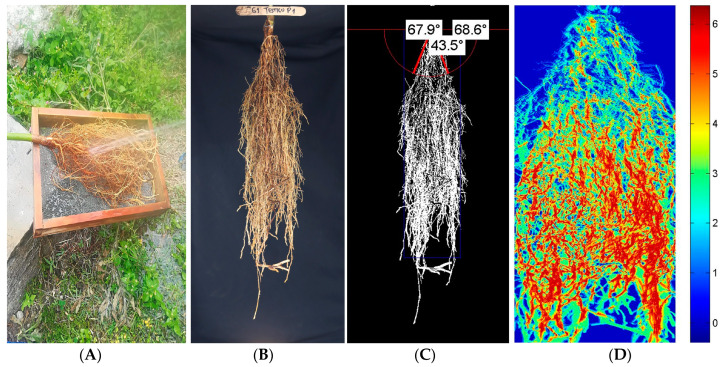
Process of avocado root conditioning for imaging: (**A**) washing, (**B**) sample before imaging, (**C**) angle opening data calculated by the software, and (**D**) density area occupied estimated by the software.

**Table 1 plants-13-02660-t001:** Mean squares for root traits of avocado plants evaluated under five water regimes.

Source of Variation	DPR (cm)	AREA (cm^2^) (×1000)	TRL (mm) (×1000)	LRA (°)	RRA (°)	RAO (°)
Repetition (Rep)	18.41	0.9	246.1	179.2	80.7	157.40
Rootstock (RS)	20.81 ^ns^	37.3 *	4772.6 ***	139.8 ^ns^	1040.7 **	508.77 **
Error A (Rep × RS)	0.34	2.0	74.4	85.72	95.1	415.81
Water regime (WR)	65.58 *	9.9 *	6920.6 ***	300.7 *	5101.0 ***	139.36 ^ns^
RS × WR	65.47 ^ns^	3.3 ^ns^	1411.0 ***	1266.9 ***	2491.0 ***	619.92 *
Error B (Rep × RS × WR)	21.43	3.3	259.2	114.14	84.8	126.37
CV, % A	21.4	15.2	13.2	14.2	15.1	22.4
CV% B	28.1	19.0	24.6	16.4	14.3	31.3

DPR: depth with a maximum proportion of roots, AREA: the size of the entire root system, TRL: total root length, LRA: left root angle, RRA: right root angle, RAO: root angle opening. *, **, and *** are significant at the 0.05, 0.01, and 0.001 probability levels, respectively. ns, non-significant at the 0.05 probability level according to the Tukey test.

**Table 2 plants-13-02660-t002:** Mean values for root traits of avocado plants under different water regimes.

Genotype	Treatments	DPR (cm)	AREA (cm^2^)	TRL (mm)	LRA (°)	RRA (°)	RAO (°)
ANRR88	T1	26.45 ^ab^	295.38 ^bc^	2353.9 ^abc^	49.03 ^c^	69.41 ^abc^	61.56 ^ab^
	T2	23.03 ^bc^	253.22 ^c^	2240.6 ^bc^	68.04 ^a^	77.60 ^a^	34.38 ^c^
	T3	31.77 ^a^	281.08 ^c^	2317.6 ^bc^	70.37 ^a^	32.83 ^d^	32.83 ^c^
	T4	15.24 ^c^	190.09 ^d^	1805.5 ^cd^	67.77 ^a^	66.05 ^bc^	46.17 ^bc^
	T5 (Control)	23.54 ^ab^	301.95 ^bc^	2644.9 ^ab^	70.56 ^a^	77.98 ^a^	31.48 ^c^
ANGI52	T1	28.77 ^ab^	290.69 ^c^	1680.4 ^d^	70.23 ^a^	72.34 ^ab^	38.43 ^c^
	T2	23.27 ^bc^	354.82 ^ab^	1560.5 ^d^	63.96 ^ab^	69.53 ^ab^	46.51 ^bc^
	T3	22.94 ^bc^	397.23 ^a^	1848.9 ^cd^	56.51 ^bc^	60.21 ^c^	63.30 ^a^
	T4	24.29 ^b^	298.46 ^bc^	1779.9 ^cd^	63.92 ^ab^	63.58 ^bc^	52.53 ^ab^
	T5 (Control)	28.63 ^ab^	357.26 ^ab^	2904.0 ^a^	68.09 ^a^	66.36 ^bc^	45.57 ^bc^
	Mean	24.95	303.14	2071.78	65.29	64.60	45.11

T1: water restriction of 50%, T2: water restriction of 25%, T3: water excess of 25%, T4: water excess of 50%, and T5: control. DPR: depth with a maximum proportion of roots, AREA: the size of the entire root system, TRL: total root length, LRA: left root angle, RRA: right root angle, and RAO: root angle opening. Different letters indicate statistical differences at a significance of *p* ≤ 0.05 according to the Tukey test.

**Table 3 plants-13-02660-t003:** Mean squares for plant height (PH), leaf area (LA), dry matter partition, and Organic Carbon (OC) of avocado plants under different water stress regimes.

Source of Variation	PH	LA (×1000)	Dry Matter (×1000)				Organic Carbon		
			Leaves	Stems	Roots	Total	Leaves	Stems	Roots
Repetition (Rep)	35.9	66.4	7.0	8.0	27.6	198.6	0.06	1.53	0.37
Rootstock (RS)	2.1 ^ns^	1679 ^ns^	10.6 ^ns^	5.3 ^ns^	332 ^ns^	602.8 ^ns^	0.10 *	615.3 ***	5.33 *
Error A (Rep × RS)	14.5	206.2	8.8	20.9	40.4	202.8	0.01	4.92	0.46
Water regime (WR)	107.9 *	2952 ***	280.8 ***	116.1 ***	202.6 *	1705.1 ***	0.50 ***	58.08 ***	3.09 ***
RS × WR	26.9 ^ns^	521.8 ^ns^	31.2 ^ns^	47.3 *	10.4 ^ns^	111.9 ^ns^	0.07 ^ns^	18.07 ***	1.15 ^ns^
Error B (Rep × RS × WR)	32.0	290.6	26.0	10.7	59.2	166.5	0.05	1.55	0.43
Mean	61.3	1.563	19.14	19.59	23.46	61.34	0.46	5.48	1.75
CV, % A	6.22	29.05	15.53	23.35	27.10	23.22	23.93	40.5	38.79
CV% B	9.22	34.48	26.68	16.74	32.81	21.04	51.67	22.78	37.69

*, and *** are significant at the 0.05, and 0.001 probability levels, respectively. ns, non-significant at the 0.05 probability level.

**Table 4 plants-13-02660-t004:** Summary of ANOVA for three vegetation indices applied to avocado plants under different water stress regimes.

Vegetation Index	Sum Square	Mean Square	F	Prob > F	CV%
NDVI	0.02297	0.00574	19.59	5.13 × 10^−16^ ***	1.09
MTCI	186.69	46.6718	23.76	1.71 × 10^−19^ ***	10.52
CI_RE_	1639.1	409.769	5.87	0.0001 **	16.72

NDVI: Normalized difference vegetation index, CI_RE_: Chlorophyll estimation red edge, and MTCI: MERIS Terrestrial chlorophyll index. **, and *** are significant at the 0.01, and 0.001 probability levels, respectively.

## Data Availability

The datasets generated and/or analyzed during the current study are available from the corresponding author on reasonable request.

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
