# Peer review of "Comparative Analysis of Water Stress Regimes in Avocado Plants during the Early Development Stage"

_plants, 2024, doi:10.3390/plants13182660_

Round 1
Reviewer 1 Report
Comments and Suggestions for Authors
The study is devoted to the topical problem of detecting water stress in avocado plants at an early stage. The materials and methods are described in detail, the introduction provides sufficient background.
Several comments on the manuscript:
1. Justify the choice of vegetation indices used in the study. It might have been better to use indices that are more sensitive to water stress, such as Water Index (WI), Moisture Stress Index (MSI) and Normalized Water Index (NWI).
2. Check Figure 2, statistically significant differences are not indicated everywhere.
3. Check the units of measurement in Figure 3.
4. The conclusions should be expanded. Specify for whom the obtained results of the study will be useful. You can also indicate directions for further research.
5. Check the list of references for compliance with the requirements.
Reviewer 2 Report
Comments and Suggestions for Authors
This manuscript introduced a very interesting study about analyzing water-stress regimes in avocado plants at early development stage. Spectral imaging, vegetation index, and image processing were used for plant phenotyping. Some suggestions are provided for improving this manuscript.
1. Line 18-91, please check these sentences.
2. Line 20, ‘Hass under four water-stress conditions …’, please kindly list these four conditions to replace this misleading expression ‘including deficit and excess’.
3. Line 27-28, ‘The REST software provided faster assessment and higher accuracy, reducing the time required for root data registration by over 50%.’ Please mention what kinds of existing methods were compared.
4. The introduction section could be improved. From the paragraph 1-6, the importance of avocado and the impact of some typical stress on its productivity and quality were introduced. However, relative studies about phenotyping of stressed avocado plants were missing. Please add this part to show the recent progress of this research topic, as well as the challenges and limitations. And then introduce what the aim and scope of this study.
5. This manuscript introduced the results in Section 2 and the materials in section 4. However, the section 2, the definitions of many abbreviations were missing (for example, ‘AREA, TRL, RRA, and RAO’, ‘T1, T2, T3, T4’). But they were found in Section 4. Therefore, it is suggested that please kindly check if this journal accept introducing the Materials first and then the Results. If yes, the Materials and Methods could be described in Section 2.
6. Table 3, ‘2952***’. ‘***’ should be used as superscript.
7. The Discussion section could be slightly simplified. Some references (e.g., published before 2019) might be removed. The comparison between this study and those recently published ones are more important. The novelty of this study should be highlighted at this part.
8. It was declared that commercial software was used for data analysis, such as ‘REST’. At the current stage, more interesting methods are applied for plant phenotyping. For examples, AI-based methods have been widely applied for drought-stressed plant analysis. I am very surprised that this ms, on such an important and dynamic subject, is based on old references (more than 5 years). And, the small number of existing researches for water stress have not all been cited and described, like for example:
[1] Poplar seedling varieties and drought stress classification based on multi-source, time-series data and deep learning
[2] Phenotyping for effects of drought levels in quinoa using remotes sensing tools
[3] UAV remote sensing phenotyping of wheat collection for response to water stress and yield prediction using machine learning
[4] Evaluating drought stress response of poplar seedlings using a proximal sensing platform via multi-parameter phenotyping and two-stage machine learning
[5] Thermal imaging: The digital eye facilitates high-throughput phenotyping traits of plant growth and stress responses
In the discussion section, please mention the possibility of using those AI-based methods for analyzing water-stress regimes in avocado plants.
Comments on the Quality of English LanguageMinor editing of English language required.
Reviewer 3 Report
Comments and Suggestions for Authors
1. Results are presented in the abstract in ambiguous language. It could be more specific.
2. Under tabe 1, full form of the terms used in the table should be mentioned.
3. In Table 2, Treatments could be described in bracket or under the table.
4. Mention what is in Y axis in figure 1
5. Which genotype is presented in figure 1?
6. In figure 2B, mention the significant difference with small letters as done in other figures.
